# Reducing the Risk of Needle Tract Seeding or Tumor Cell Dissemination during Needle Biopsy Procedures

**DOI:** 10.3390/cancers16020317

**Published:** 2024-01-11

**Authors:** Dennis R. Holmes

**Affiliations:** Adventist Health Glendale, 1505 Wilson Terrace, Suite 370, Glendale, CA 91206, USA; drholmesmd50@gmail.com

**Keywords:** needle biopsy, breast cancer, cryoablation, seeding, recurrence, fine needle aspiration, coaxial

## Abstract

**Simple Summary:**

Many women fear that breast needle biopsies increase the risk of cancer spread due to tumor cell displacement. Although evidence suggests minimal impact on breast cancer loco-regional recurrence or distant metastasis, technical modifications of needle biopsy procedures, including the use of cryoablation, can reduce the potential risks of breast needle-biopsy-induced cancer cell displacement to reduce patient fears of breast needle biopsy procedures.

**Abstract:**

Purpose: Many women fear that breast needle biopsies increase the risk of cancer spread. The purpose of this review article is to discuss the breast cancer literature regarding the risk of needle-biopsy-induced cancer cell displacement and its impact on local and regional recurrence and breast cancer survival. Methods: A literature review is performed to discuss the risks and mitigation of needle-biopsy-induced cancer cell displacement. Results: Needle-biopsy-induced cancer cell displacement is a common event. The risk is influenced by the biopsy technique and the breast cancer type. Evidence suggests that the risk of needle-biopsy-induced cancer cell displacement may potentially increase the odds of local recurrence but has no impact on regional recurrence and long-term survival. Conclusions: Technical modifications of needle biopsy procedures can reduce the risk of breast needle-biopsy-induced cancer cell displacement and potentially reduce the risk of local recurrence, especially in patients for whom whole breast radiation is to be omitted.

## 1. Introduction

Minimally invasive breast biopsy or diagnostic needle biopsy is currently the standard of care for obtaining an initial tissue diagnosis of a breast abnormality. The emergence of this standard was driven in part by the common desire to reduce the psychological and esthetic burden, morbidity, and healthcare expense of routine diagnostic breast surgical excisional biopsies of breast abnormalities, the majority of which are likely to be benign [1,2,3]. Consequently, the National Consortium of Breast Centers, the American Society of Breast Surgeons, and the American College of Surgeons quality metrics have established diagnostic needle biopsy (either core needle biopsy or fine needle aspiration) as the most appropriate initial diagnostic approach for breast abnormalities to allow for appropriate selection and pre-operative treatment planning for the minority of patients for whom lesion surgical resection may be appropriate, including those with malignant, high-risk, or discordant pathology or symptomatic lesions [3,4,5,6]. Although diagnostic breast excisional biopsy may sometimes be warranted, there is a consensus that diagnostic excisional biopsy should be reserved for the few situations where a needle biopsy might be infeasible due to technical reasons (e.g., the inability to obtain an adequate sample) or patient safety concerns (e.g., underlying coagulopathy) [7]. 

For patients ultimately diagnosed with breast cancer, diagnostic needle biopsy facilitates the multidisciplinary management of breast cancer in ways that directly impact the quality of breast cancer care as well as the patient’s short- and long-term outcomes. Diagnostic needle biopsy facilitates thorough pre-treatment counseling and the selection of appropriate surgical options (e.g., lumpectomy, mastectomy, and lymph node surgery), including referral to plastic surgery for the coordination of simultaneous breast reconstruction or oncoplastic surgery [3]. It provides adequate tissue for ancillary testing (e.g., tumor biomarkers) and facilitates decision-making regarding the benefits for neoadjuvant chemotherapy vs. adjuvant systemic therapy [8]. Diagnostic needle biopsy reduces the risk of positive surgical margins and the need for re-excision, which, in turn, reduces associated surgical morbidity (e.g., hematoma, infection, poor cosmesis) [1,9]. A pre-operative diagnosis of cancer also facilitates second opinions, which might make available to the patient treatment options that they might not have been previously considered, including clinical trials. 

Despite the importance of diagnostic needle biopsy in the management of breast cancer, many patients are reluctant to undergo the procedure for fear that penetration of the tumor with a biopsy needle might itself cause the cancer to spread—a concern which leads some patients to defer needle biopsy procedures for weeks of months. On the other hand, there is compelling evidence that a ≥3-month delay between abnormal imaging and a positive needle biopsy may lead to a 20% relative reduction in overall survival [10]. Thus, ironically, delays in performing a diagnostic needle biopsy can directly increase the risk of cancer cell spread and death among the very women who most fear a diagnostic needle biopsy. 

Patient concerns about the potential for cancer cell dissemination appear to be partly justified. Indeed, a large body of data show needle tract seeding to be a relatively common event after a breast diagnostic needle biopsy, and the needle-biopsy-induced intravascular dissemination of tumor cells has been observed in animal and human studies [11,12,13,14,15,16,17,18,19,20,21,22]. Nonetheless, studies assessing the long-term implications of breast needle-biopsy-associated tumor cell displacement demonstrate no statistically significant adverse impact on the rate of breast cancer local recurrence, distant metastasis, or overall survival [11,12,23].

Despite the absence of harm, patient anxiety about the theoretical risk of cancer spread could be a barrier to quality breast care. Therefore, to reduce patient anxiety, it is appropriate for healthcare providers to employ practical mitigation or prevention measures to reduce the potential for needle-biopsy-associated cancer cell displacement or dissemination. Although no mitigation technique is fail-safe, technical modifications of the needle biopsy procedure can be utilized to encourage a greater proportion of reluctant patients to safely proceed with a recommended diagnostic needle biopsy.

## 2. Mitigation or Prevention Measures

**Epinephrine-containing anesthetic field block.** Common to each of the strategies listed below is the use of a local anesthetic field block containing epinephrine (concentration of 1:100,000–1:200,000 or 5–10 μg/mL). A field block is performed by injecting a local anesthetic solution along all of the margins (near, far, superficial, deep, medial and lateral, and superior and inferior) of the lesion that is to be biopsied. The primary role of epinephrine in the local anesthetic mixture is to cause the vasoconstriction of nearby arteries and veins to restrict blood flow into and out of the area, extend the duration of the pain-relieving anesthetic, and theoretically decrease the potential for the dissemination of cancer cells into the bloodstream [24,25]. These effects last up to 6 h.

**Use of a coaxial or introducer.** A coaxial or introducer is a large-diameter needle that can be inserted into the breast through which a small-diameter biopsy needle can be inserted to obtain one or more biopsy samples. Coaxial use decreases procedure time, reduces tissue trauma, and reduces tissue contact by isolating the needle tract for insertion and removal of a needle biopsy device multiple times, which theoretically reduces the risk of needle tract seeding. Although there are no breast cancer studies evaluating the impact of coaxial use on the rate of needle tract seeding, data from hepatocellular cancer demonstrate a lower risk of needle tract seeding with the use of a coaxial [26,27]. Therefore, coaxial use is a reasonable consideration for reducing the risk of breast biopsy needle tract seeding.

**Fine needle aspiration.** Ultrasound-guided or palpation-guided fine needle aspiration (FNA) utilizes a small needle to collect individual or clusters of cells. In general, FNA is performed with a 21–27-gauge needle, a fraction of the diameter of biopsy devices that are typically used for a core needle biopsy (CNB). By collecting a smaller sample of cells, FNA reduces disruption of the tumor mass, decreases bleeding, and potentially lowers the risk of cancer spread.

While FNA has its advantages, several limitations are noteworthy. Firstly, while FNA cytology can distinguish benign from malignant lesions, it cannot distinguish invasive cancer from in situ cancer due to its inability to assess tissue architecture in a cytology specimen. Secondly, FNA carries a greater risk of obtaining a non-diagnostic sample due to the relatively small quantity of cells collected with each aspiration. Fortunately, the risk of inadequate sampling can be minimized by having a cytopathologist available at the time of the FNA to rapidly assess sample quality. If the initial sample is non-diagnostic, additional samples can be immediately obtained until an adequate sample has been verified.

**Core needle biopsy with limited tissue sampling.** To ensure adequate tissue sampling, many physicians utilize large-gauge biopsy devices to obtain multiple tissue samples. Often, 5–10 large biopsy specimens are obtained with vacuum-assisted devices, and 3–5 specimens are commonly obtained with automated gun (spring-loaded) devices, though fewer samples are often sufficient. Although there are no direct correlations between core biopsy instrument diameter and the rate of needle tract seeding for vacuum-assisted and spring-loaded devices, technical differences in the biopsy procedure do affect the risk of needle tract seeding. Whereas vacuum-assisted devices are capable of obtaining multiple samples with only a single insertion into the breast, spring-loaded devices require device insertion and removal multiple times to retrieve the specimen and re-load the device after each sampling. Consequently, limiting the number of spring-loaded biopsy needle passes to three or less would reasonably be expected to reduce the risk of cancer cell displacement, especially when combined with a coaxial and an epinephrine-containing anesthetic field block.

**Cryoablation-assisted needle biopsy.** Percutaneous cryoablation is emerging as a minimally invasive alternative to lumpectomy for the management of early-stage breast cancer. Cryoablation utilizes a specialized needle (a cryoprobe) and liquid nitrogen or argon gas to achieve targeted tissue ablation using ultra-low temperatures. Although the typical aim of cryoablation is therapeutic (i.e., complete tumor eradication), cryoablation is playing an increasingly important role in the diagnostic phase of care to facilitate tissue sampling and to minimize the needle tract seeding of cancer cells. Pulmonary medicine provides a perfect example of this, where cryoprobe transbronchial lung biopsy and cryoablation-assisted lung biopsy improve tissue sampling while decreasing complications in the diagnosis of benign and malignant lung conditions [28,29].

Cryoablation-assisted needle biopsy (CAB) directly addresses the risk of needle tract seeding by enabling ablation of the needle tract immediately after needle biopsy. CAB can be utilized in patients who are willing to undergo lumpectomy, mastectomy, or neoadjuvant chemotherapy, but are unwilling to undergo a diagnostic needle biopsy. CAB can also be utilized to facilitate the needle biopsy of tumors that are poor candidates for therapeutic cryoablation due to large tumor size, locally advanced breast cancer, multifocal/multicentric breast cancer, or other reasons.

With the patient positioned supine or supine oblique, a needle biopsy trajectory is chosen that can be used for both the needle biopsy device and the cryoprobe, which ideally is a trajectory that is parallel to the chest wall through the longest horizontal axis of the tumor. Local anesthetic containing epinephrine is then injected as a field block. After creating a dermotomy, the coaxial and needle biopsy device are inserted under ultrasound guidance to the center of the tumor. Ultrasound-guided needle biopsies are obtained, followed by removal of the coaxial. Immediately after withdrawal of the coaxial, the cryoprobe is then inserted under ultrasound guidance through the same needle tract until the tip of the cryoprobe enters the mass. Cryoablation is then initiated with freezing and ablation of the needle tract within and adjacent to the mass, with the goal of ablating the needle tract and a surrounding 2–4 mm radial tissue margin. The duration of each ablation is determined by the cryoablation treatment algorithm, which is usually calculated based on the diameter of the tissue to be ablated. For example, if the diameter of the coaxial/needle tract is 3 mm, then the minimum target diameter of ablation would be 7 mm (3 mm + 2 mm + 2 mm) or greater. A wider cryoablation zone of necrosis would be needed if biopsies were obtained on multiple planes or levels. After completing ablation of the intra-tumor needle tract, the cryoprobe is warmed, and then, withdrawn up to 4 cm (or up to the level of the subcutaneous tissue), at which time a single freeze cycle is repeated to ablate the remaining, unablated segment of the needle tract.

**Cryoablation-guided needle biopsy.** Cryoablation-guided needle biopsy (CGB) is a suitable solution for an individual who has elected to undergo breast cryoablation but is reluctant to undergo a diagnostic needle biopsy due to a fear of needle tract seeding or cancer cell dissemination. CGB makes it possible for the needle biopsy to be performed at the same time as the cryoablation procedure, which eliminates a delay between specimen collection and the tumor ablation. CBG rapidly kills cancer cells that might have been displaced into the needle tract. By enabling rapid freezing of the entire tumor and surrounding margin, CGB obstructs arteries, veins, and lymphatic vessels that are feeding the mass, thereby preventing the shedding of cells into the bloodstream or lymphatic system.

The initial steps of CGB and CAB are similar. A needle biopsy trajectory is chosen that is optimal for the cryoprobe as well as the needle biopsy, which is ideally a trajectory parallel to the chest wall through the longest horizontal axis of the tumor. Images are taken and measurements are obtained to select the appropriate cryoablation treatment algorithm. A field block is then created using an epinephrine-containing local anesthetic solution. After making a dermotomy, the cryoprobe is inserted under ultrasound guidance through the center of the visible lesion. With an assistant holding the cryoprobe securely in place, a second dermotomy is created adjacent to the first dermotomy, and the coaxial and needle biopsy devices are inserted under ultrasound guidance directly in line with the cryoprobe, such that the cryoprobe and biopsy needles are positioned on adjacent planes. Ideally, the axis of the biopsy needle would lie immediately anterior or superficial to the cryoprobe. However, the bulky profile of the handpieces of most biopsy instruments and cryoprobes prevents perfect parallel placement of both instruments (Figure 1). Consequently, the pragmatic approach typically calls for the biopsy needle to be inserted at a downward angle immediately adjacent to the cryoprobe, creating a shallow “X”-like orientation of the devices (Figure 1 and Figure 2). With the cryoprobe remaining in position, up to three good quality biopsy specimens are obtained via the coaxial. With both instruments held in place, cryoablation is then initiated before removing the coaxial and biopsy needle containing the final specimen. Iceball growth is then monitored via ultrasound until the biopsy needle is no longer visible within the mass. At this point, the coaxial and needle biopsy device are immediately withdrawn from the breast before both are solidly frozen in place. The cryoablation procedure is continued according to the cryoablation treatment algorithm, with the goal of ablating the mass plus a ≥5 mm surrounding normal tissue ablation margin. The cryoprobe is then withdrawn along the cryoprobe tract up to 4 cm or until the final 4 cm of the cryoprobe remains within the breast, assuming that the length of the freeze zone is 4 cm. Cryoablation is then initiated with a single freeze cycle and monitored via ultrasound until the iceball extends >5 mm beyond the axis of the adjacent biopsy needle tract.

Although it is unlikely that a momentary freeze will alter the tissue architecture or the immunohistochemical properties of the specimen, a reasonable precaution is to label the final specimen separately to distinguish it from any samples that might have been removed prior to freezing. However, leaving the biopsy needle and specimen within the mass until the very end of the mass cryoablation could compromise specimen histology such that it might be unsuitable for histological assessment [30].

## 3. Discussion

Although most breast needle biopsies yield a benign diagnosis, there is substantial concern amongst the public that traumatic penetration of cancer with a needle biopsy device causes the shedding of cancer cells into the needle tract or bloodstream, which may increase the risk of local recurrence or distant metastasis. As a result, some patients with suspicious breast abnormalities are reluctant to undergo a diagnostic needle biopsy. In reality, invasive cancers develop the ability to spread from the moment they become vascularized (~2 mm), which is typically years before they reach the average dimension (14 mm) typically detected by annual mammograms [23,31]. As a result, 3–6% of women diagnosed with breast cancer today are found to have de novo stage IV disease, meaning that clinically detectable distant metastasis had already been established well before the diagnostic needle biopsy was performed [32]. Furthermore, positive axillary nodes are detected in up to 27% of patients with breast cancers 2 cm or smaller, in up to 62% of patients with breast cancers between 2 cm and 5 cm, and in up to 78% of patients with breast cancers >5 cm, indicating that disease has already been established in the lymph nodes before any needle biopsies have been performed [11]. Considering that most cancers exhibit growth and possible spread months or years before they are detected by mammography, ultrasound, breast MRI, or examination, there is sufficient reason to doubt the significance of a single needle biopsy event to a patient’s risk of local recurrence or death.

Many publications have examined the implications of needle-biopsy-associated needle tract seeding and tumor cell dissemination in various malignancies. In the breast cancer literature, Diaz et al. published the most comprehensive analysis of needle tract seeding in a study of 352 women in which needle tract seeding was detected in 32% of participants (n = 114) [12]. When the results were stratified by the interval between needle biopsy and surgery, the authors found needle tract seeding in 42% of patients who underwent surgery 15 days after needle biopsy, compared to only 15% of patients who underwent surgery > 28 days after the needle biopsy. The actual quantity of detected cells also diminished over the same time interval. In another study, Uematsu et al. detected positive cytology from 75% of washings obtained from 18-gauge spring-loaded biopsy devices immediately after withdrawal from the breast and before specimen removal—essential at time zero [13]. Considered together, these studies suggest that most displaced cancer cells fail to survive for even one month, either due to tumor cell apoptosis or natural immune clearance. In addition, Diaz et al. detected no significant difference in the overall rate of needle tract seeding when comparing spring-loaded devices and vacuum-assisted devices, although a larger quantity of displaced cells was generally observed with spring-loaded devices. Tissue histology also impacted the rate of needle tract seeding, with tract seeding observed in 40% of patients with invasive ductal carcinoma compared to 15% of patients with invasive lobular carcinoma and 16% of patients with mixed invasive ductal and lobular cancer. There was no statistical correlation between needle tract seeding and tumor size, tumor grade, or the presence of lymphatic space invasion.

There is a dichotomy in the literature related to the rate of needle tract seeding. Similar to Diaz, multiple studies report a 17–38% rate of needle tract seeding associated with 14-gauge-automated and vacuum-assisted needle biopsy devices, while other studies describe needle tract seeding as a rare event (0–2%) [12,13,14,15,16,17,18,19]. The wide variance between these studies likely relates to differences in needle biopsy devices, the number of biopsies obtained, the number of needle passes performed, the interval between needle biopsy and surgery, and tumor histology (ductal vs. lobular).

Another major area of concern regarding needle biopsy procedures is the shedding of viable cancer cells into the lymphatic system and bloodstream. The risk of lymphatic shedding was examined by Hansen et al., who reported a significantly higher rate of sentinel lymph node metastasis after FNA and CNB compared to excisional biopsy [33]. However, two important study design flaws challenged the validity of their findings: the authors failed to account for tumor histology, which is known to be strongly linked to the risk of tumor cell displacement, and they failed to exclude patients treated with neoadjuvant chemotherapy who likely carried a higher baseline risk of positive sentinel nodes due to larger tumor size. Nonetheless, after accounting for those factors, Chagpar and Peters-Engl detected no significant difference in the rate of positive sentinel nodes upon comparing FNA, CNB, excisional biopsy, and incisional biopsy among women with clinically node-negative breast cancer [34,35].

Although the needle-biopsy-induced intravascular dissemination of cancer cells has been described in animal studies, Hu et al. conducted one of the only human breast cancer studies evaluating the impact of FNA on the detection of circulating tumor cells (CTC) [20,21,22]. In their study, 20 women with malignant tumors (Stage I–IV) underwent arm blood draws obtained immediately before and immediately after a 21-gauge needle was used to perform three consecutive breast FNAs. Using RT-PCR to identify three invasive cancer-specific markers, Hu et al. identified CTC in the post-biopsy blood in four (20%) cancer patients who were formerly CTC-negative. While this finding validates the concern that needle biopsy is capable of shedding cancer cells into the bloodstream, the viability of the CTC remains unknown. More significantly, the small sample size should only characterize these findings as “hypothesis-generating”, for which a larger study of CNB should be conducted in early-stage breast cancer given the widespread use of CNB and the importance of the issue to breast cancer patients and their providers.

Some publications suggest that the risks of needle tract seeding and cell dissemination are related to the diameter or gauge of the needle biopsy device [36,37,38]. However, due to the heterogeneous design of most studies and their general lack of controls, it is difficult to prove a definitive association between biopsy device diameter and tumor cell displacement. Confounding the analysis is the inconsistent or unspecified use of coaxials or the lack of specification regarding the number of specimens obtained, both of which may impact the risk of tract seeding. Furthermore, the method of sampling may be more important than the device gauge used. Apart from requiring fewer passes, it is also hypothesized that the suction feature of vacuum-assisted devices might extract cells that could otherwise disseminate or seed the needle tract.

Notwithstanding the lower level of available evidence, there is sufficient reason to conclude that needle tract seeding and tumor cell shedding are commonly associated with needle biopsy procedures. However, the more important question is whether these phenomena translate into an increased risk of local recurrence and/or death. To answer this question, Leibens et al. performed a systematic analysis of 3 prospective and 12 retrospective studies published before 2009 [39]. Although needle tract seeding was reported in 22% of patients overall, the analysis found no increased risk of local recurrence or death after needle biopsy.

The preponderance of evidence shows no significant association between needle biopsy and the risk of local recurrence or death. Nevertheless, the lack of randomized or controlled studies, the heterogeneous design of available studies, and the general failure to stratify results by tissue histology, coaxial use, and the number passes/samples leave some room for doubt about the safety of needle biopsy procedures. For this reason, several modifications of the needle biopsy procedure have been presented to increase the safety of needle biopsy procedures. Included among these are two novel procedures (cryoablation-assisted biopsy and cryoablation-guided biopsy), which combine percutaneous cryoablation of the needle tract with or without the tumor and surrounding blood vessels. Available evidence shows 100% successful ablation of tumors <1 cm in diameter, and between 92–98% of tumors 2 cm or smaller [40]. Although the ability to ablate peripheral vessels with CGB depends on the tumor size and the maximum volume of ablation achieved, there is clear evidence that CAB is capable of completely ablating biopsy needle tracts, all of which are <1 cm in diameter.

## 4. Conclusions

Concerns about needle tract seeding and tumor cell dissemination lead some women with suspected breast cancer to avoid undergoing recommended diagnostic needle biopsies. Although the evidence shows no significant association between needle biopsy and the risk of local recurrence or death, the quality of the evidence leaves some room for doubt about the safety of needle biopsy procedures. By utilizing one or more of these mitigation strategies, providers will be better equipped to encourage reluctant patients to undergo a diagnostic needle biopsy with confidence that the needle biopsy procedure itself will not compromise their cancer control and long-term survival.

## Figures and Tables

**Figure 1 cancers-16-00317-f001:**
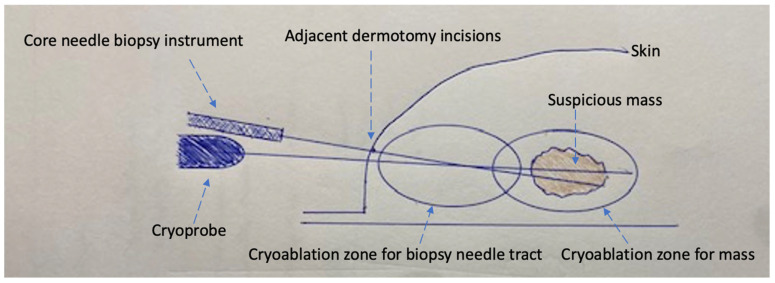
Schematic of cryoablation-guided needle biopsy procedure showing cryoprobe traversing mass, needle biopsy device within mass (coaxial not shown), two adjacent dermotomy incisions, cryoablation zone for the suspicious mass, and additional cryoablation zone for the needle biopsy tract.

**Figure 2 cancers-16-00317-f002:**
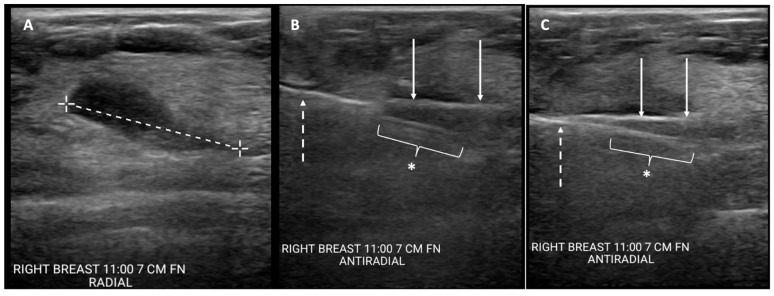
Cryoablation-guided needle biopsy procedure. Ultrasound images showing (**A**) cancer prior to insertion of devices (broken line indicates longest lesion dimension); (**B**) cryoprobe inserted through mass (solid arrows) and spring-loaded core needle biopsy device (broken arrow) with aperture open (*), pre-fire; and (**C**) cryoprobe inserted through mass (solid arrows) with core needle biopsy device (broken arrow) with aperture closed (*), post-fire.

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
