# Peer review of "Reducing the Risk of Needle Tract Seeding or Tumor Cell Dissemination during Needle Biopsy Procedures"

_cancers, 2024, doi:10.3390/cancers16020317_

Round 1

Reviewer 1 Report

Comments and Suggestions for Authors

The concise review article deals with a common medical problem and a question of significant concerns to the patients, namely if and how needle biopsies may cause dissemination of cancer cells. This is a long lasting issue for both medical profession and patients. Although the question is supported by only reference (Ref-1), this is almost a daily question and practitioner face. Thus, it is an important area to address.

The proposed way to mitigate the challenge  are well documented and the discussion is appropriate. 

One minor suggestion is that it would be useful to suggest how the proposed measures and practice be followed up, in order that the proposal is appropriately assessed and instituted at practice.

Author Response

Thanks for your comments and suggest.

I think it would be difficult to monitor the outcome of the proposed techniques other than with routine follow-up mammogram and ultrasound.  Based on the preponderance of the evidence, I would conclude that needle tract seeding does not significantly alter the event rate. Therefore, it would be difficult to show how any intervention would improve on that.  However, since some patients are skeptical of the evidence, taking additional mitigation measures might to helpful in getting some to undergo needle biopsies.

Reviewer 2 Report

Comments and Suggestions for Authors

The study is interesting an it can contribute to better diagnosis of patients with breast cancer. Although needle-biopsy induced cancer cell displacement is a common event, technical modifications of needle biopsy procedure can  lower the risk and potential consequences of cancer cell displacement, and reduce patient anxiety about the procedure.

 Follow comments and suggestions for authors:

How are patients advised about the risk of procedures?

What are the criteria adopted to recommend a needle biopsy?

For diagnosis, compare traditional method com Needle Biopsy Procedures

Line 45 – 47 – cite references for animal and human studies regarding procedures and tumor dissemination

In the introduction, it would be interesting to add more references to improve this review's approach, it should be done.

Author Response

Thanks for reviewing the manuscript and providing comments.

Follow comments and suggestions for authors:

How are patients advised about the risk of procedures?  There is the usual informed consent process as for any invasive procedure. The risks are provided in written form and discussed verbally by the radiologist. This may or not may not include a discussion of the minimal risk of spread from the needle biopsy.

What are the criteria adopted to recommend a needle biopsy?  I have added to the manuscript the indications for a needle biopsy.

For diagnosis, compare traditional method com Needle Biopsy Procedures.  I have added a discussion of this to the opening paragraphs of the manuscript.

Line 45 – 47 – cite references for animal and human studies regarding procedures and tumor dissemination.   Citation added.

In the introduction, it would be interesting to add more references to improve this review's approach, it should be done.  Additional references have been added.

Round 2

Reviewer 2 Report

Comments and Suggestions for Authors

The questions and inconistencies identified in the first review were satisfactorily addressed.